# A genetic modifier links *integrin α5* to the phenotypic variation in *fibronectin 1a* mutant zebrafish

Samuel J. Capon[1,2,3]*, Anastasia Maroufidou[1,2], McKenna Feltes[4], Yanli Xu[1,2,3], Darpan Kaur Matharoo[1,2], Dörthe Jülich[5], Scott A. Holley[5], Steven A. Farber[4], Didier Y. R. Stainier[1,2,3]*

1 Department of Developmental Genetics, Max Planck Institute for Heart and Lung Research, Bad Nauheim, Germany, 2 German Centre for Cardiovascular Research (DZHK), Partner Site Rhine-Main, Bad Nauheim, Germany, 3 Cardio-Pulmonary Institute (CPI), Max Planck Institute for Heart and Lung Research, Bad Nauheim, Germany, 4 Department of Biology, Johns Hopkins University, Baltimore, Maryland, United States of America, 5 Department of Molecular, Cellular and Developmental Biology, Yale University, New Haven, Connecticut, United States of America

* samuel.capon@mpi-bn.mpg.de (SJC); didier.stainier@mpi-bn.mpg.de (DYRS)

## Abstract

Phenotypic variation is often observed in individuals with the same mutation. However, the mechanisms that contribute to this variation remain largely unknown. Fibronectin mutants in both mouse and zebrafish fail to form a functional cardiovascular system, although the penetrance and expressivity of this phenotype vary depending on the genetic background. Here we investigate the variation of the zebrafish *natter* phenotype, which is caused by a nonsense mutation in *fibronectin 1a* (*fn1a*). *natter*/*fn1a* mutants exhibit incompletely penetrant cardia bifida, a phenotype caused by the failure of cardiac progenitors to migrate to the midline. To examine whether this variation is related to the nonsense mutation, we first generated a large deletion in *fn1a* that removes the proximal promoter and first 17 exons. Characterisation of this allele found that mutants display variable cardiac phenotypes indistinguishable from those observed in *natter*/*fn1a* mutants. As phenotypic variation is often associated with changes in paralogous gene expression, we next examined the expression of the *fn1a* paralogue, *fn1b*, and observed its upregulation specifically in the *natter*/*fn1a* mutants that exhibit a severe phenotype. However, overexpression and double mutant analyses suggest that *fn1b* expression levels do not modulate the *natter*/*fn1a* mutant phenotype. During these studies, we observed a small proportion of *natter*/*fn1a* mutants with a wild-type (WT)-like phenotype. Selectively raising WT looking mutant larvae increased the proportion of *natter*/*fn1a* mutants displaying the WT-like phenotype from 1.7% to 38.6% in just three generations, indicating the selection of a genetic modifier of the mutant phenotype. We mapped this modifier to the *integrin alpha 5* (*itgα5*) locus through whole-genome sequencing. Furthermore, we found that manipulating *itgα5* expression influenced the severity of the *fn1a* mutant

**Data availability statement:** The whole-genome sequencing data reported in this paper have been deposited in the European Nucleotide Archive (ENA) under accession number PRJEB89164. All other relevant data are within the manuscript and its supporting information files.

**Funding:** This research was supported by funds from the Max Planck Society as well as awards from the European Research Council (ERC) under the European Union's research and innovation programs (AdG 694455-ZMOD and AdG 101021349-TAaGC) awarded to D.Y.R.S.; and NIH grants F32GM144223 to M.F., R01DK093399 to S.A.F., and R35GM148348 to S.A.H. The funders had no role in study design, data collection and analysis, decision to publish or preparation of the manuscript.

**Competing interests:** The authors have declared that no competing interests exist.

phenotype, and that the variance in *itga5* expression was increased in *fn1a* mutants exhibiting a severe phenotype. Taken together, these results indicate that *itga5* modifies the *fn1a* mutant phenotype.

## Author summary

Recent studies have identified apparently healthy individuals with mutations that would be expected to lead to disease. Why these individuals are disease resistant is not understood. Studies in zebrafish and other organisms have revealed that environmental factors, stochastic fluctuations in gene expression, and genetic modifiers can contribute to such phenotypic variation. However, the precise mechanisms underlying phenotypic variation are likely unique to each mutation. Here, we present an analysis of zebrafish *natter/fibronectin 1a* (*fn1a*) mutants. These mutants display variable cardiac, brain, and somite defects. *Fibronectin* mutants in mouse display similar phenotypes, and compelling evidence that they are affected by a genetic modifier has been reported. We find that the zebrafish *natter/fn1a* mutant phenotype is dependent on developmental temperature, a genetic modifier, and an age-dependent parental factor. Using a simple breeding strategy, we selected for mutants that display milder phenotypes. Coupling selection with whole-genome sequencing, we mapped a genetic modifier to the *integrin alpha 5* (*itga5*) gene, the primary receptor for Fibronectin. Further analyses of *itga5* expression in *fn1a* mutants and manipulating *itga5* expression in *fn1a* mutants indicate that *itga5* modifies the *natter/fn1a* mutant phenotype. The results presented here provide new insights into the mechanisms regulating phenotypic variation in *natter/fn1a* mutants.

## Introduction

Phenotypic variability, the propensity of a phenotype to vary, and phenotypic variation, the actual differences observed [1], are commonly reported in the characterisation of mutant phenotypes. However, the underlying mechanisms are poorly understood. Phenotypic variation can manifest itself as incomplete penetrance, where a proportion of mutants do not display the phenotype, as well as variable expressivity, where the strength of the phenotype varies between individual mutants [2]. In the most extreme cases, identical mutations can vary in their effect, from being lethal in one individual to having no observable effect in another. In line with this idea, recent studies have identified mutations that would be expected to give rise to severe and fully-penetrant disorders in apparently healthy humans [3,4]. How these individuals, nicknamed the superheroes of disease resistance [5], achieve this remarkable feat is unknown. Additional analyses have estimated that every human genome harbours approximately 100 loss-of-function (LoF) mutations leading to the complete inactivation of approximately 20 genes [4, 6], suggesting that disease

resistance may be more widespread than previously appreciated. Consistent with this thinking, it was recently reported that the mean penetrance of over 5000 dominant, LoF, pathogenic variants is as low as 6.9% [7]. These findings highlight the importance of studying the mechanisms underlying phenotypic variation in order to better understand the genotype–phenotype relationships.

The mechanisms contributing to phenotypic variation are likely diverse and potentially specific to each pathogenic variant; and therefore, detailed studies in suitable model systems are needed. Zebrafish have been used extensively to study early development owing to their high fecundity, facile genetics, optical transparency, and *ex utero* development. These characteristics, as well as the heterogeneity of the zebrafish genome [8,9], also lend themselves to studying phenotypic variation. Previous work in zebrafish and other organisms have revealed that genetic modifiers, stochastic fluctuations in gene expression, and environmental factors contribute to phenotypic variation [10–15]. A common theme emerging from this work is that variable paralogous gene expression is linked to phenotypic variation. An intuitive interpretation is that upregulation of functionally related genes compensates for the mutated gene [16,17].

The early embryonic heart is composed of two cell types, the inner endocardial cells and the outer myocardial cells, with the myocardium powering the rhythmic contractions of the heart [18]. During vertebrate development, cardiac progenitors are bilaterally localised in the lateral margins of the anterior lateral plate mesoderm and migrate to the midline to form the linear heart tube [19–23]. Fibronectin is a large extracellular matrix protein involved in a number of developmental and regenerative processes [24–34]. In zebrafish, Fibronectin is encoded by the *fibronectin 1a* (*fn1a*) and *fibronectin 1b* (*fn1b*) genes owing to the teleost specific whole-genome duplication [35,36]. *fn1a* is expressed in endocardial precursors during cardiac progenitor cell migration in zebrafish [37,38] and is required for myocardial cell migration to the midline, with *natter*/*fn1a^tl43c^* mutants displaying a cardia bifida phenotype [25,28]. However, the penetrance of the cardia bifida phenotype in *natter*/*fn1a^tl43c^* mutants is variable, with developmental temperature and genetic background altering the phenotype [25,28]. No defects have been observed in *fn1b* mutants, suggesting that it is dispensable for development [39]. However, *fn1a*; *fn1b* double mutants display a more severe phenotype than *fn1a* single mutants, suggesting that *fn1b* can, at least partially, compensate for *fn1a* function and vice versa [39]. Cells interact with Fibronectin through Integrin receptors on their cell surface, and the Itgα5/Itgβ1 heterodimer is the primary Fibronectin receptor [40]. The *integrin alpha 5* (*itgα5*) gene is expressed in both the developing endocardium and myocardium in zebrafish [41]. Although *itgα5* mutants do not display cardia bifida, *itgα5*; *itgα4* double mutants do, with variable penetrance and expressivity [41], similar to *natter*/*fn1a^tl43c^* mutants. Altogether, these studies indicate that, in zebrafish, Fibronectin signalling through integrin receptors is required for myocardial cell migration to the midline.

Similar findings have been reported in rodent models: *Fibronectin 1* (*Fn1*) is expressed in endocardial progenitors in rat embryos [42] and mouse mutants display a variable cardia bifida phenotype [24,43,44]. Moreover, *Fn1* mutant mice on the 129/Sv genetic background display a severe phenotype, and yet the same mutant allele on a C57BL/6J background displays only mild defects [43]. Further examination of the cardiac phenotype in mouse identified a locus on chromosome 4 linked to the severity of the myocardial migration defect [44], suggestive of a genetic modifier of the *Fn1* mutant phenotype. However, the precise variant and underlying mechanisms remain unknown.

Based on all these findings, we hypothesised that a genetic modifier influences the penetrance and expressivity of the cardia bifida phenotype in zebrafish *natter*/*fn1a^tl43c^* mutants. Here, we first report that the variation of the *fn1a* mutant phenotype is regulated by developmental temperature, genetic modifiers, and an age-dependent parental factor. We then used a simple breeding strategy to select for genetic modifiers of the *fn1a* mutant phenotype, and whole-genome sequencing analysis led to the identification of a modifier in the *itgα5* locus. Additional data further indicate that *itgα5* modifies the *fn1a* mutant phenotype, with increased variance in *itgα5* expression observed in *fn1a* mutants with a more severe phenotype.

## Results

### *natter/fn1a^tl43c* mutants display a variable cardiac phenotype

To investigate the variable phenotype in *natter/fn1a^tl43c* mutants, we first performed *in situ* hybridisation (ISH) on 24 hours post fertilisation (hpf) embryos from single pair heterozygous intercrosses with a *myl7* probe to label the cardiomyocytes. Wild-type (WT) siblings form a linear heart tube as expected, whereas *natter/fn1a^tl43c* mutants display variable cardiac phenotypes (Fig 1A). The most severe phenotype observed was cardia bifida, whereby the myocardial progenitors fail to migrate to the midline. However, we also observed mutants where the myocardial progenitors had fused at the midline. We further classified these mutants according to whether the fused progenitors formed a symmetric or asymmetric structure. The symmetric phenotype more closely resembles the cardiac disc structure observed in WT embryos at an earlier stage, suggesting that the migration and cohesion of myocardial progenitors is less affected in these embryos, although further analysis would be needed to test this hypothesis. Surprisingly, 6/41 mutant embryos formed a linear heart tube indistinguishable from that of WT siblings (Fig 1A and 1B). Comparison of the cardiac phenotype in four *natter/fn1a^tl43c* clutches revealed significant variation in both penetrance and expressivity (Fig 1B).

As this analysis identified mutant embryos with a linear heart tube, we next examined whether mutant larvae could develop a fully functional heart. To this aim, we intercrossed heterozygous *natter/fn1a^tl43c* zebrafish and raised the embryos at 28.5°C. Gross phenotypic analysis of mutant larvae at 100 hpf identified significant phenotypic variation (Fig 1C). Most *natter/fn1a^tl43c* mutants developed a severe phenotype consisting of pronounced pericardial oedema, a small head, and disorganised trunk muscle fibres (white arrows, Fig 1C). However, some mutants developed a milder phenotype where the only distinguishing feature was a less pronounced pericardial oedema (white arrowhead, Fig 1C). Again, we noted that a small proportion of mutants displayed a phenotype that was nearly identical to that of WT siblings. We were able to raise most of these WT looking mutants to adulthood under standard conditions and without any intervention. These mutant adults did not display any overt defects and were fertile. To further analyse these phenotypes, we imaged the hearts of mutants displaying the WT-like phenotype as well as those displaying a mild or a severe phenotype, as defined by the extent of pericardial oedema, from a *Tg(myl7:EGFP); natter/fn1a^tl43c* mutant incross at 100 hpf. For comparison, we also imaged WT siblings from a *natter/fn1a^tl43c* heterozygous intercross at the same stage. The expressivity of the cardiac phenotypes varied considerably (Fig 1D). Most WT looking mutants displayed subtle morphological defects in the heart, specifically in the outer curvature of the ventricle, as well as in the size and shape of the atrium. Mild mutants exhibited more pronounced cardiac defects, while severe mutants developed the stringy heart phenotype associated with pronounced pericardial oedema. Quantification of the 100 hpf mutant phenotype from three clutches revealed significant variation in both penetrance and expressivity (Fig 1E), as previously noted for the 24 hpf phenotype.

### *fn1a^bns692* mutants also display a variable cardiac phenotype

*natter/fn1a^tl43c* mutants harbour a nonsense mutation in the second exon of the *fn1a* gene, which is predicted to lead to nonsense-mediated decay. Recent results from our lab and others have found that mutations that lead to premature termination codons can activate genetic compensation via a protein-independent mechanism known as transcriptional adaptation (TA) [45–47]. In an attempt to avoid TA in this study, we generated a large deletion in the *fn1a* locus that removes the proximal promoter as well as the first 17 exons (Figs 2A, and S1), hereafter referred to as the *fn1a^bns692* allele. The expression of *fn1a* in *fn1a^bns692* mutants relative to WT siblings is just 0.5 ± 0.07% (S2A–C Fig), compared with *natter/fn1a^tl43c* mutants where the relative expression is 25.4 ± 1.1% of WT siblings (S2G–I Fig). However, although we were unable to detect *fn1a* expression in 5/12 *bns692* mutant larvae examined using a primer pair inside the deleted region, and the remaining 7/12 mutant larvae displayed Cq values higher than those of no RT controls (S2D Fig), two primer pairs outside the deleted region consistently detected low levels of transcripts from the intact exons (S2E and S2F Fig). These results suggest that the *fn1a^bns692* allele is not a completely RNA-less allele and could therefore still trigger TA. However,

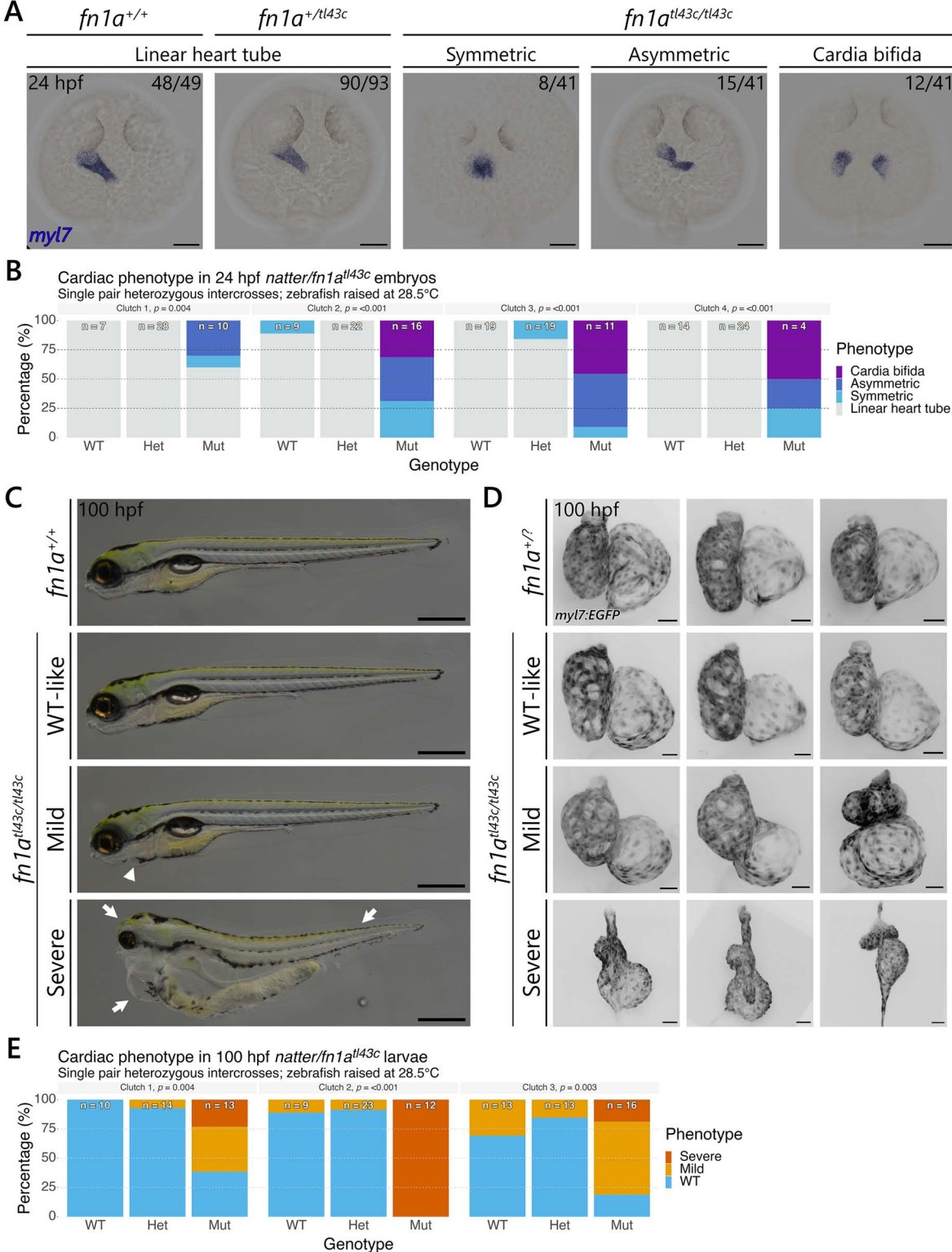

**Fig 1. natter/fn1a^tl43c mutants display a variable cardiac phenotype. (A)** *In situ* hybridisation with the myocardial marker *myl7* was performed on 24 hpf *natter/fn1a^tl43c* mutant embryos and WT siblings from four single pair *natter/fn1a^tl43c* heterozygous intercrosses. The proportion of embryos matching the image shown is indicated in the top right corner of each image. The remaining 6/41 mutant embryos displayed a linear heart tube phenotype as

noted in **(B)**. Dorsal views are shown with anterior to the top. **(B)** Quantification of the 24 hpf myocardial migration phenotype in *natter/fn1a^tl43c^* mutants and WT siblings shown in **(A)**. **(C)** Representative images of a 100 hpf homozygous WT sibling as well as a WT-like, a mild, and a severe *natter/fn1a^tl43c^* mutant larva from a single pair *natter/fn1a^tl43c^* heterozygous intercross. Lateral views are shown with anterior to the left. The mild mutant phenotype is distinguishable by mild pericardial oedema (white arrowhead). The severe mutant phenotype is distinguishable by a small head, pronounced pericardial oedema, and disorganised trunk muscle fibres (white arrows). **(D)** Confocal images of three hearts representative of each phenotype were selected from imaging 12 WT-like, 6 mild, and 6 severe *Tg(myl7:EGFP)* mutants from a single pair *natter/fn1a^tl43c^* mutant incross at 100 hpf. For the *fn1a^+/+^* and *fn1a^+/tl43c^* siblings, referred to collectively as *fn1a^+/?^*, three representative images were selected from imaging 12 WT *fn1a^+/?^* siblings from a single pair *natter/fn1a^tl43c^* heterozygous intercross at 100 hpf. **(E)** Quantification of the 100 hpf cardiac phenotype in *natter/fn1a^tl43c^* mutants and WT siblings from three single pair *natter/fn1a^tl43c^* heterozygous intercrosses. The numbers on each bar in **(B)** and **(E)** indicate the total number of larvae assessed. All p-values were calculated using chi-squared tests. Scale bars, 100 μm in **(A)** and **(D)**, and 1 mm in **(C)**.

any protein produced from this allele would lack the signal peptide as well as the N-terminal 70 kDa fragment that plays important roles in cell and protein interactions [40], likely resulting in a non-functional protein if translated. We next examined the phenotype in *fn1a^bns692^* mutants by ISH at 24 hpf with a *myl7* probe as well as by gross phenotypic analysis at 100 hpf, and found that *fn1a^bns692^* mutants phenocopy *natter/fn1a^tl43c^* mutants, including the variable penetrance and expressivity (Fig 2B–D).

### *fn1b* expression levels correlate with the severity of the *fn1a* mutant cardiac phenotype

Recent studies have shown that paralogous gene expression is linked to phenotypic variation in zebrafish and other species [10–15]. Zebrafish possess two fibronectin genes, *fn1a* and *fn1b*. To examine *fn1b* expression in *fn1a* mutants, we first performed RT-qPCR for *fn1b* in *fn1a* mutants as well as WT siblings collected from *natter/fn1a^tl43c^* and *fn1a^bns692^* heterozygous intercrosses at the protruding mouth stage (72 hpf). We observed a clear upregulation of *fn1b* in *fn1a* mutants with the severe phenotype for both the *natter/tl43c* and *bns692* alleles relative to their homozygous WT siblings (Fig 3A). Notably, we never observed *fn1b* upregulation in mutant larvae with a WT-like phenotype, or even a mild phenotype. To examine the spatial pattern of this upregulation, we performed ISH with an *fn1b* probe in protruding mouth stage *fn1a^bns692^* mutants and homozygous WT siblings (Fig 3B and 3C). This analysis revealed, in a subset of mutants, a clear upregulation of *fn1b* expression over the yolk, where *fn1b* is expressed in WT siblings, as well as ectopic expression in the cardiac region, where *fn1b* expression was not observed in WT siblings. To further examine *fn1b* expression in the heart, we dissected *fn1a^bns692^* mutant larvae at the protruding mouth stage and pooled the hearts of mutant larvae displaying the WT-like and mild phenotypes together, and hearts of mutant larvae displaying the severe phenotype in a separate pool. We also collected the remaining bodies from these dissections for comparison. RT-qPCR analysis revealed similar *fn1b* upregulation in both the hearts and bodies of severe mutants from three independent single pair *fn1a^bns692^* homozygous incrosses (S3F Fig). These results confirm that *fn1b* is upregulated in the heart of severe mutants.

To investigate whether the upregulation of *fn1b* drives the severe mutant phenotype, we next injected *fn1b* mRNA into 1-cell stage embryos from single pair *natter/fn1a^tl43c^* and *fn1a^bns692^* homozygous mutant incrosses. For comparison, we also injected *fn1a* mRNA. Injection of either *fn1b* or *fn1a* significantly rescued the mutant phenotype in both the *natter/tl43c* and *bns692* alleles (Figs 3D and S3A–C). Furthermore, injecting *fn1a* mRNA also normalised the upregulation of *fn1b* observed in *fn1a^bns692^* mutants (S3D and S3E Fig), suggesting that the upregulation of *fn1b* in *fn1a* mutants is a result of the loss of Fn1a protein. We then crossed the *fn1a^bns692^* allele to the *fn1b^ya14Tg^* allele [34,39] and examined the phenotype of double mutant larvae and siblings from *fn1a^bns692^; fn1b^ya14Tg^* double heterozygous intercrosses. This analysis found that even a single copy of the *fn1b* mutant allele significantly increased the severity of the *fn1a* mutant phenotype (Fig 3E), while two copies of the *fn1b* mutant allele did not further increase this phenotypic severity. Taken together, these results indicate that while *fn1b* is specifically upregulated in *fn1a* mutants with the severe phenotype, this upregulation is not the cause of the increased severity.

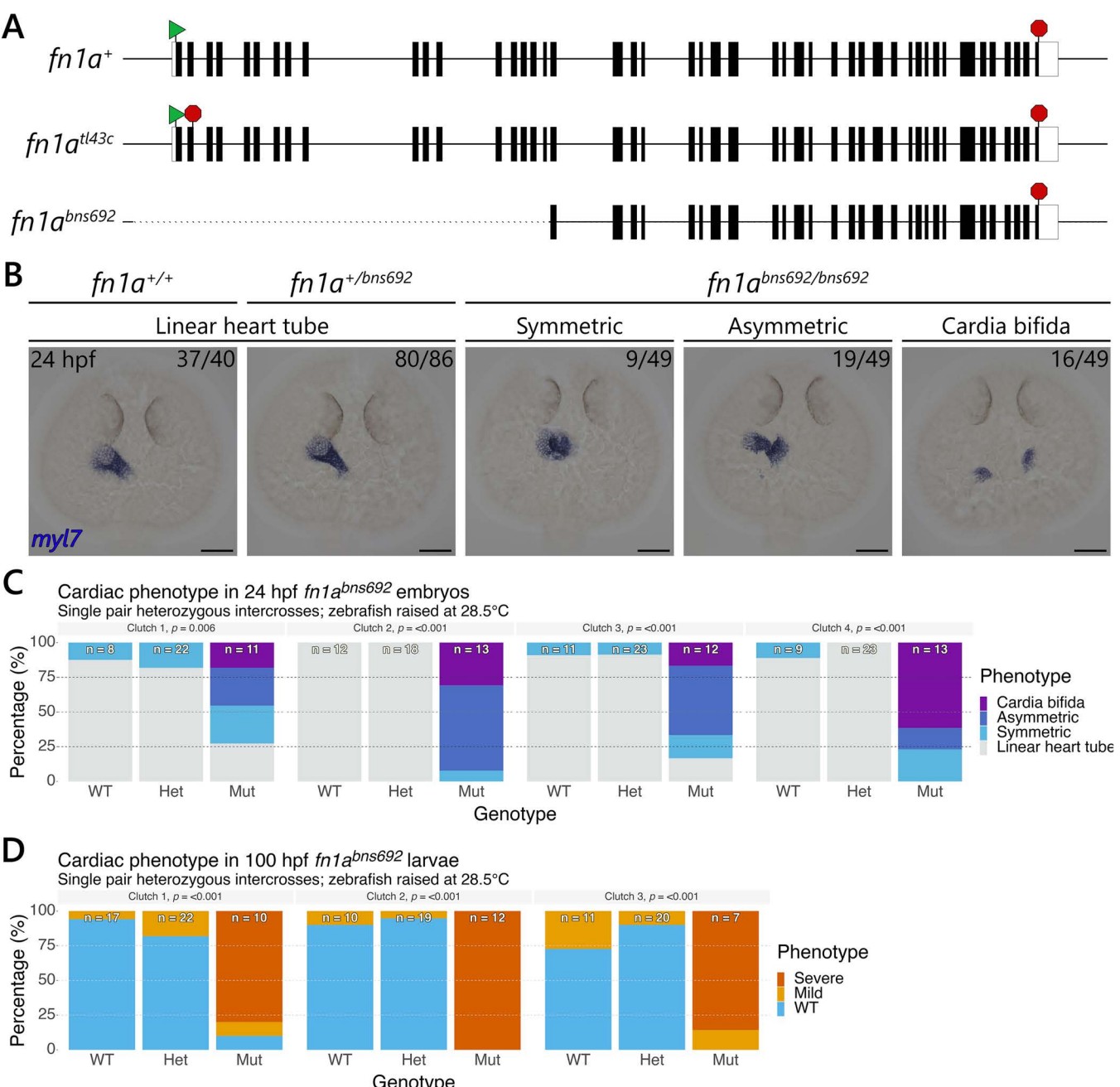

**Fig 2. *fn1a^bns692* mutants also display a variable cardiac phenotype. (A)** Schematic showing the WT, *natter/tl43c*, and *bns692 fn1a* alleles. Exons (n = 46) are indicated by black boxes. The start codons are represented by a green triangle and the stop codons by a red octagon. **(B)** *In situ* hybridisation with the myocardial marker *myl7* was performed on 24 hpf *fn1a^bns692* mutant embryos and WT siblings from four single pair *fn1a^bns692* heterozygous intercrosses. The proportion of embryos matching the image shown is indicated in the top right corner of each image. The remaining 5/49 mutant embryos displayed a linear heart tube phenotype as noted in **(C)**. Dorsal views are shown with anterior to the top. **(C)** Quantification of the 24 hpf myocardial migration phenotype in *fn1a^bns692* mutants and WT siblings shown in **(B)**. **(D)** Quantification of the 100 hpf cardiac phenotype in *fn1a^bns692* mutants and WT siblings from three single pair *fn1a^bns692* heterozygous intercrosses. The numbers on each bar in **(C)** and **(D)** indicate the total number of larvae assessed. All p-values were calculated using chi-squared tests. Scale bars, 100 μm.

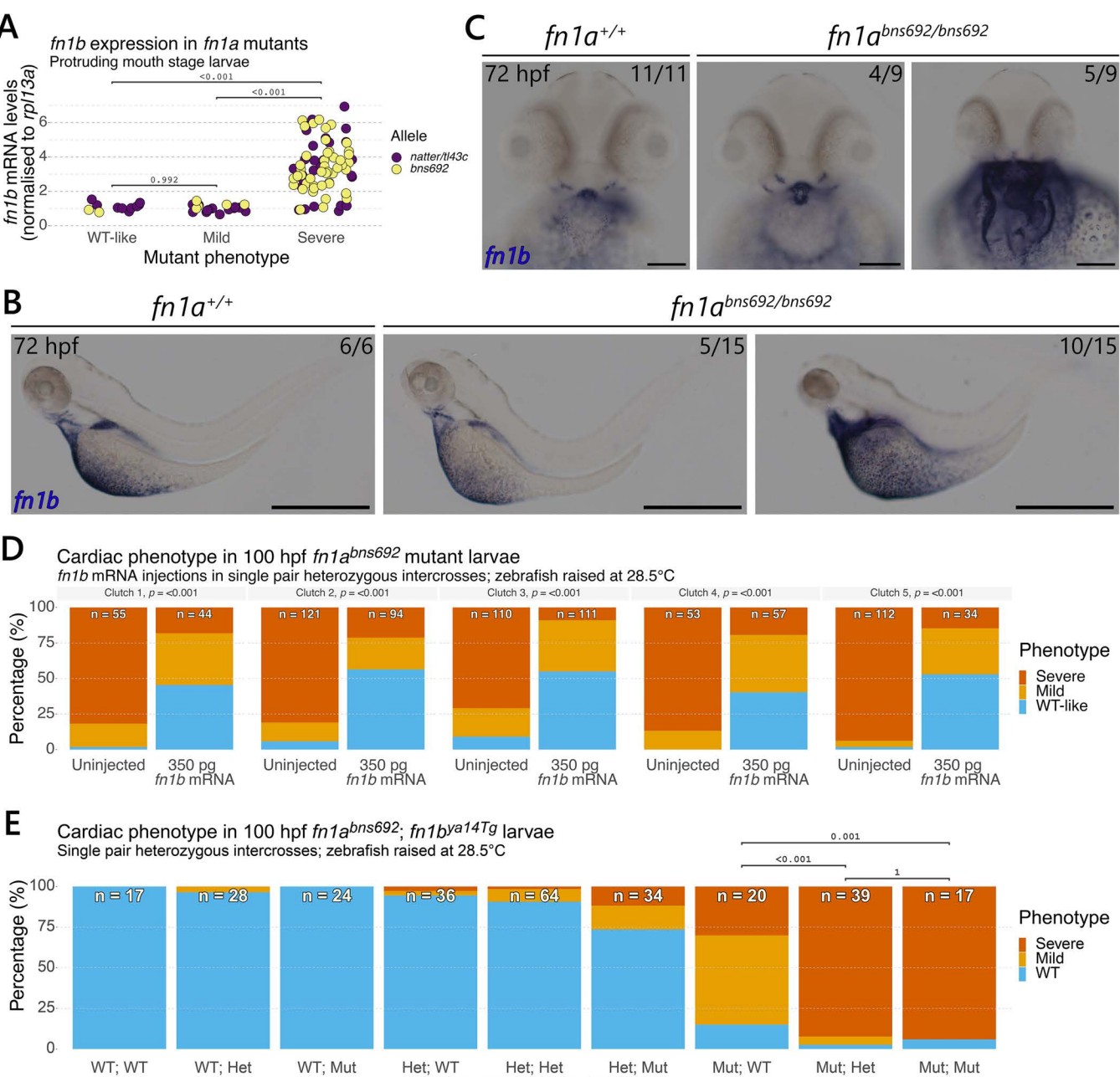

**Fig 3. *fn1b* expression levels correlate with the severity of the *natter/fn1a^tl43c^* and *fn1a^bns692^* mutant cardiac phenotype. (A)** *fn1b* mRNA levels were analysed by RT-qPCR in individual 72 hpf protruding mouth stage larvae collected from single pair *natter/fn1a^tl43c^* and *fn1a^bns692^* heterozygous intercrosses. Each data point represents an individual larva. Expression is relative to homozygous WT siblings in each clutch. Eight clutches were examined for each allele from experiments described in S2 and S4 Figs. **(B, C)** *In situ* hybridisation for *fn1b* was performed on 72 hpf larvae from two single pair *fn1a^bns692^* heterozygous intercrosses. The proportion of larvae matching the image shown is indicated in the top right corner of each image. **(B)** Lateral views with anterior to the left. **(C)** Ventral views with anterior to the top. **(D)** Quantification of the 100 hpf cardiac phenotype in mutant larvae injected with 350 pg *fn1b* mRNA and uninjected controls from five single pair *fn1a^bns692^* mutant incrosses. **(E)** Quantification of the 100 hpf cardiac phenotype in *fn1a^bns692^; fn1b^ya14Tg^* mutants and siblings from three single pair *fn1a^bns692^; fn1b^ya14Tg^* double heterozygous intercrosses. The numbers on each bar in **(D)** and **(E)** indicate the total number of larvae assessed. For **(A)**, p-values were calculated by one-way ANOVA with Tukey's post-hoc test for multiple comparisons; for **(D)**, p-values were calculated with chi-squared tests; and for **(E)**, p-values were calculated by Fisher's exact test with Bonferroni correction for multiple comparisons. Scale bars, 1 mm in **(B)** and 100 μm in **(C)**.

## The phenotypic variation of *fn1a* mutants is determined by developmental temperature, a genetic modifier, and an age-dependent parental factor

The penetrance of the cardia bifida phenotype in *natter*/*fn1a^{tl43c}* mutants has been reported to be dependent on developmental temperature and genetic background [25,28]. To investigate the effect of temperature, we conducted single pair *natter*/*fn1a^{tl43c}* heterozygous intercrosses and raised the embryos at 28.5°C. At the bud stage (10 hpf), we shifted one dish of embryos from each clutch to 22°C, another to 32°C, and left a third dish at 28.5°C. We phenotyped the larvae once they reached the equivalent of a 100 hpf stage larva raised at 28.5°C, and collected them for genotyping. In line with the previous reports, we observed that mutant larvae displayed a more severe phenotype with increasing temperature in all clutches examined for both the *natter*/*fn1a^{tl43c}* and *fn1a^{bns692}* alleles (S4A and S4B Fig). However, this increase in severity reached statistical significance in only one of the three clutches examined, likely due to the low number of mutants obtained.

We next examined the hypothesis that the temperature sensitivity is due to increased mRNA levels of the mutated gene at lower temperatures (e.g., from increased stability) by repeating the temperature shift experiment and collecting larvae at the protruding mouth stage (i.e., 72 hpf when raised at 28.5°C) for RT-qPCR analysis. While we observed statistically significant changes in *fn1a* mRNA levels relative to WT siblings for both the *natter*/*fn1a^{tl43c}* and *fn1a^{bns692}* alleles (S4C–H Fig and S1 Table), these changes were small and inconsistent between alleles, and thus unlikely to explain the phenotypic variation (20.2%±0.4% at 22°C, 19.7%±0.9% at 28.5°C, 17.6%±0.7% at 32°C, for *natter*/*fn1a^{tl43c}* mutants; 0.4%±0.04% at 22°C, 0.6%±0.04% at 28.5°C, 0.9%±0.1% at 32°C, for *fn1a^{bns692}* mutants).

We next tested whether we could breed for the WT-like mutant phenotype using a simple survival strategy (Fig 4A). We initially took advantage of the temperature sensitivity to generate F1 populations of mutant adults for each allele. F1 mutant adults were then incrossed and the progeny raised at 28.5°C to minimise the effect of temperature in all subsequent experiments. At 100 hpf, the larvae were phenotyped according to our previously established criteria (Fig 1). We did not observe differences between the zygotic and maternal zygotic phenotypes; however, we did observe a small proportion of WT looking mutant larvae at 100 hpf, as noted previously (Figs 1E and 2D). We raised these WT looking F2 mutant larvae in our aquarium under standard conditions to generate F2 mutant adults. Incrossing F2 mutants and raising the progeny at 28.5°C increased the proportion of mutant larvae displaying the WT-like phenotype from 1.7% in F1 to 8.9% in F2 *natter*/*fn1a^{tl43c}* incrosses (Fig 4A'), and from 6.1% in F1 to 6.9% in F2 *fn1a^{bns692}* incrosses (Fig 4A"). Repeating this experiment with F3 mutants increased the proportion of mutant larvae displaying the WT-like phenotype to 25.6% in *natter*/*fn1a^{tl43c}* incrosses and 16.8% in *fn1a^{bns692}* incrosses. We also applied additional selection pressure to a separate population of F3 *natter*/*fn1a^{tl43c}* mutant larvae by shifting the temperature to 32°C during development and raising the WT looking mutant larvae as before. Incrossing these F3 [32°C] mutant adults further increased the proportion of mutant larvae displaying the WT-like phenotype to 38.6% (Fig 4A'). These data indicate that one can rapidly select for genetic modifiers of the *fn1a* mutant phenotype using a simple survival strategy.

In addition, while incrossing F3 *natter*/*fn1a^{tl43c}* mutants, we noted that the penetrance and expressivity of the mutant phenotype varied from week-to-week in repeated incrosses of the same mating pair (S5 Fig). With increasing age, five out of the eleven analysed pairs displayed a consistent decline in the proportion of mutant larvae displaying the WT-like phenotype. Although the other six pairs did not follow a consistent decline, they did follow the same general trend with more severe phenotypic outcomes observed in the progeny of older zebrafish. Taken together, these results reveal that the variation of the *fn1a* mutant phenotype is complex, being determined by a combination of developmental temperature, genetic modifiers, and age-dependent parental factors.

## A genetic modifier links *itga5* to the phenotypic variation in *fn1a* mutants

To identify genetic modifiers of the *fn1a* mutant phenotype, we first incrossed a single pair of the F2 *natter*/*fn1a^{tl43c}* mutant zebrafish described in Fig 4A' and raised the embryos at room temperature. The WT looking larvae from this incross

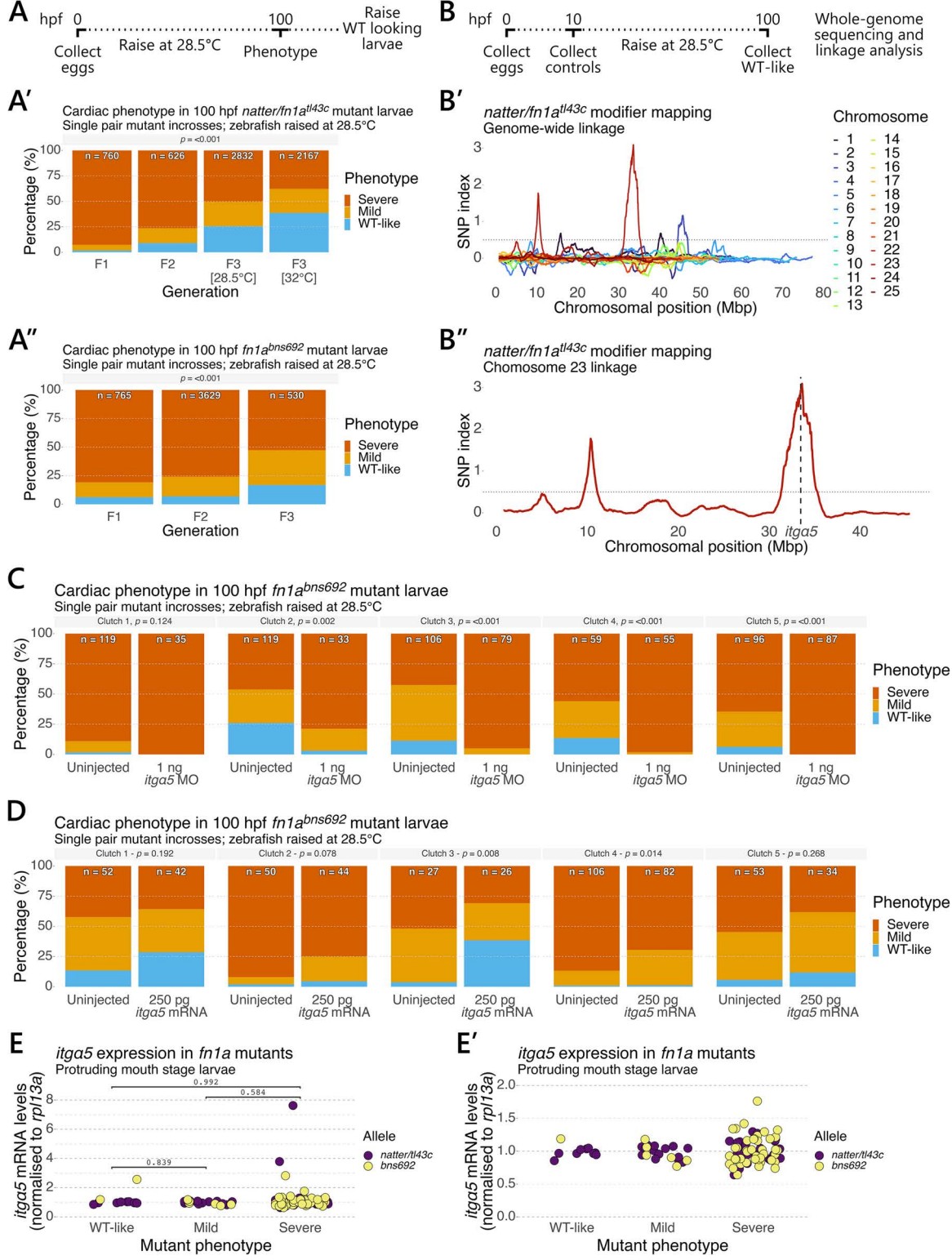

**Fig 4. A genetic modifier links *itga5* to the phenotypic variation in *fn1a* mutants.** (A) Outline of the breeding scheme to select *fn1a* mutants that display milder phenotypes. Multi-pair mutant incrosses were performed and the embryos raised at 28.5°C. Larvae were phenotyped at 100 hpf and the WT looking larvae raised to establish the next generation. Quantification of the 100 hpf cardiac phenotype in *natter/fn1a*<sup>tl43c</sup> **(A')** and *fn1a*<sup>bns692</sup> **(A")** mutants from single pair mutant incrosses over 3 generations. *natter/fn1a*<sup>tl43c</sup> F3 [28.5°C] and F3 [32°C] zebrafish were raised during development at

28.5°C and 32°C, respectively. (B) Outline of the breeding strategy used to map the genetic modifier in *natter/fn1a^tl43c* mutants, see materials and methods for a detailed description. A single dominant linkage peak was identified on chromosome 23 **(B')** mapping to the *itga5* locus **(B")**. 4 additional peaks with a SNP index ≥0.5 were identified on chromosomes 1, 3, and 23 **(B')**. (C) Quantification of the 100 hpf cardiac phenotype in uninjected and 1 ng *itga5* MO injected mutant larvae from five single pair *fn1a^bns692* mutant incrosses. (D) Quantification of the 100 hpf cardiac phenotype in uninjected and 250 pg *itga5* mRNA injected mutant larvae from five single pair *fn1a^bns692* mutant incrosses. The numbers on each bar in **(C)** and **(D)** indicate the total number of larvae assessed. (E) *itga5* mRNA levels were analysed by RT-qPCR in individual 72 hpf protruding mouth stage larvae collected from single pair *natter/fn1a^tl43c* and *fn1a^bns692* heterozygous intercrosses. Each data point represents an individual larva. Expression is relative to homozygous WT siblings in each clutch. Eight clutches were examined for each allele from experiments described in S2 and S4 Figs. **(E')** The data from **(E)** are displayed with a shortened Y-axis to show the increased variance in *itga5* mRNA levels in severe mutants. For **(C)** and (D), p-values were calculated using a chi-squared test; and for **(A)** p-values were calculated by one-way ANOVA with Tukey's post-hoc test for multiple comparisons.

were raised under standard conditions to generate a mapping population of 18 F3 *natter/fn1a^tl43c* mutant adult zebrafish descendent from a single F2 mating pair, thereby minimising the number of alleles in this population. This F3 population was separated by sex and placed in groups of 3–4 zebrafish resulting in two groups of males and three groups of females. Each group was housed separately over the course of the experiment. A round-robin breeding scheme was used whereby each group of males was set up with each group of females in multi-pair matings over three successive weeks to ensure that the embryos were collected from as many different animals as possible. From each clutch, we collected 100 embryos at 10 hpf as controls, as well as all of the mutant larvae displaying the WT-like phenotype at 100 hpf (Fig 4B). DNA was extracted from these populations and equal proportions from each clutch were combined to generate 1) a control sample that could serve to estimate allele frequencies in this mutant population, and 2) a WT-like sample to examine which variants were enriched in mutants displaying the WT-like phenotype. We then used the WheresWalker mapping pipeline [48] to identify the loci under selection in the mutants displaying the WT-like phenotype.

This analysis identified five peaks with a SNP index ≥0.5, indicating linkage to the WT-like mutant phenotype (Fig 4B'). Two peaks were found on chromosome 1 (14.44 - 25.81 Mbp, maximum SNP index = 0.68; 38.58 - 53.21 Mbp, maximum SNP index = 0.68); a single peak on chromosome 3 (43.82 - 47.49 Mbp, maximum SNP index = 1.16); and two peaks on chromosome 23 (6.65 - 13.96 Mbp, maximum SNP index = 1.78; 29.69 - 36.77 Mbp, maximum SNP index = 3.09). The linkage peak on chromosome 23 between 29.69 - 36.77 Mbp had the highest SNP index and was prioritised for further analysis. We examined the genes in this region and found that the peak was centred on the *itga5* locus (Fig 4B"), and that there were 75,138 unique variants in this region (S2 Table). We next used Ensembl Variant Effect Predictor [49] to predict the consequences of each variant. This analysis found that 74,132 of these variants were associated exclusively with non-coding sequences; 638 variants were associated with both non-coding and coding sequences; and just 368 variants were associated exclusively with coding sequences. Most variants were biallelic, although we also detected some loci with 3 variants, suggesting that there were at least 3 alleles in our mapping population. We focused on the coding variants in *itga5* given its relationship with Fn1 signalling. 16 variants were identified in the *itga5* coding sequence; however, just 3 of these lead to non-synonymous changes (Q619R; H848Q; T923S) and none are predicted to significantly impact protein function based on their SIFT score [50]. Each of these 3 non-synonymous variants was detected in the control sample, with the WT-like mutant sample homozygous for the reference allele. The allele frequency was similar for each of these variants suggesting that it is possible, although unlikely, that all 3 of these variants were in the same allele.

This result prompted us to test the hypothesis that *itga5* expression modifies the *fn1a* mutant phenotype. To this end, we utilised a previously published *itga5* morpholino oligonucleotide (MO) [51]. WT larvae injected with 5 ng of *itga5* MO failed to inflate their swim bladder, developed craniofacial defects and pericardial oedema (white arrowheads, S6A Fig), and displayed defects in somitogenesis (S6B Fig), all consistent with the published mutant phenotypes [51]. By contrast, injecting just 1 ng of *itga5* MO did not lead to observable defects in the majority of larvae, although a small proportion developed U-shaped somites similar to those observed in *itga5* mutants (white arrows, S6B Fig). As injecting 1 ng of *itga5* MO had little effect on cardiac development, we repeated these injections in embryos from single pair *fn1a^bns692* homozygous mutant

incrosses to examine whether a reduction in Itgα5 expression could modify the *fn1a* mutant phenotype. We observed a statistically significant increase in the severity of the mutant phenotype in 4/5 clutches examined, with the severity of the *fn1a* mutant phenotype also increasing in the 5th clutch without reaching statistical significance (Fig 4C).

We next injected 250 pg of *itgα5* mRNA into embryos from single pair *fn1a*[bns692] homozygous mutant incrosses to examine whether *itgα5* overexpression could modify the *fn1a* mutant phenotype. We observed a statistically significant decrease in the severity of the mutant phenotype in 3/5 clutches examined, with the severity of the *fn1a* mutant phenotype also decreasing in the remaining clutches without reaching statistical significance (Fig 4D). In addition, we examined *itgα5* mRNA levels by RT-qPCR in the same samples used to examine *fn1b* expression in Fig 3A. We did not observe a significant change in overall expression levels of *itgα5* in *fn1a* mutants with the severe phenotype (Fig 4E). However, further examination of these data revealed increased variance in the mutants with the severe phenotype relative to the WT-like and mild phenotypes (*itgα5* expression variance in WT looking mutants = 0.12; in mild mutants = 0.01; and in severe mutants = 0.81) (Fig 4E'). Taken together, these results indicate that *itgα5* modifies the *fn1a* mutant phenotype, with increased variance in *itgα5* mRNA levels associated with increased severity.

## Discussion

Our analysis of *fn1a* mutant zebrafish found that the phenotypic variation observed in these animals is determined by developmental temperature, genetic modifiers, and age-dependent parental factors. To gain a mechanistic understanding of how these factors influence the *fn1a* mutant phenotype, we focused on the role of genetic modifiers. Linkage analysis identified a strong peak centred on the *itgα5* locus, and data from partial knockdown of Itgα5 as well as *itgα5* mRNA overexpression indicate that *itgα5* expression modifies the *fn1a* mutant phenotype. Furthermore, our expression analysis revealed that increased variance in *itgα5* mRNA levels is associated with the more severe *fn1a* mutant phenotype.

To our knowledge, the mechanisms that regulate temperature sensitive phenotypes in zebrafish remain largely unknown. Interestingly, the *fn1a*[bns692] allele appeared to be less sensitive to a change in temperature from 28.5°C to 32°C in all three clutches. These results suggest that the temperature sensitivity of these alleles may be influenced by the specific mutation, although further experiments are needed with larger sample sizes to address this observation. We did not observe consistent changes in *fn1a* mRNA levels for either allele over the temperatures examined in our experiments, suggesting that some other parameter(s) causes the temperature sensitivity of *fn1a* mutants, e.g., changes in the expression or function of other genes and/or proteins. Heat-shock proteins are a promising candidate in this context. One hypothesis suggests that these proteins are diverted from their usual role in buffering against cryptic genetic variation [52, 53] to manage increased protein misfolding during the heat-shock response [54], thereby revealing cryptic variants that alter the mutant phenotype. However, it is also possible that heat-shock proteins modify the *fn1a* mutant phenotype by regulating the expression and/or function of genes/proteins involved in the compensatory response directly, or even that heat-shock proteins do not play a role in the temperature sensitivity of the mutant phenotype. Interestingly, it has also been found that in zebrafish, a hypomorphic *s1pr2* mutant as well as *gata5* and *mixl1* morphants display temperature sensitive cardia bifida phenotypes associated with changes in the expression of genes encoding extracellular matrix components, including *fn1a* [55]. These results suggest that myocardial cell migration in zebrafish is an inherently temperature dependent process, although the exact mechanism remains unclear.

Similarly, while we found that the *fn1a* phenotype is altered by age-related parental factors, little is known about the impact of parental age on mutant phenotypes in zebrafish. An age-dependent phenotype has been reported for the *janus* mutant [56, 57], and it is likely that many more remain unreported. Interestingly, the *janus* mutant phenotype is also temperature dependent, suggesting a possible mechanistic link between temperature sensitivity and the impact of parental age on mutant phenotypes. One hypothesis is that the maternally derived yolk may be affected by age. The yolk-syncytial layer (YSL) has previously been found to influence myocardial cell migration, at least in part through the regulation of Fibronectin expression and extracellular matrix assembly [58–60]. Furthermore, small heat-shock proteins are expressed

in the YSL [61] and YSL specific depletion of these proteins leads to a variable cardia bifida phenotype [61, 62]. Despite these observations, further experiments are clearly needed to address how parental age, and developmental temperature, modulate the *fn1a* mutant phenotype.

An emerging concept in the study of phenotypic variation is that it is intertwined with the regulation of paralogous gene expression [10–15]. An intuitive interpretation of these results is that upregulation of functionally related gene(s) compensates for the mutated gene [16, 17]. Surprisingly, however, we found that upregulation of *fn1b* in *fn1a* mutants is strongly correlated with detrimental outcomes, at least in larvae. We found that overexpression of *fn1a* abolishes this upregulation of *fn1b*, which we observed in most of the severe *fn1a* mutant larvae at 72 hpf. These results suggest that this late *fn1b* upregulation is most likely the result of the loss of Fn1a protein. Furthermore, we found that *fn1b* overexpression also rescues the *fn1a* mutant phenotype, and that just one copy of a *fn1b* mutant allele is sufficient to significantly increase the severity of the *fn1a* mutant phenotype. Taken together, these results suggest that the upregulation of endogenous *fn1b* we observed in *fn1a* mutant larvae is a consequence, rather than a cause, of the severe phenotype.

A study in mouse identified a genetic modifier of the *Fn1* mutant phenotype on chromosome 4 [44]; however, *Itgα5* is located on chromosome 15 (at approximately 103 Mbp). In this study [44], the distal-most marker used to examine linkage on chromosome 15 is located at 88,250,634 Mbp (Sophie Astrof, personal communication). Thus, an association with *Itgα5* in mouse may have been missed due to a lack of sufficient coverage, although it is of course possible that the mouse and zebrafish modifiers are different. It will be interesting to reexamine the mouse linkage data in light of our results.

We observed, in our linkage analysis, 4 additional peaks with a SNP index ≥ 0.5 on chromosomes 1, 3, and 23. Analysing the synteny between these loci and the mouse modifier locus on chromosome 4 using the NCBI Comparative Genome Viewer [63] does not suggest that any of these zebrafish loci correspond to the mouse modifier. However, while analysing the SNP frequency in our data, we found that the modifier allele in the *itgα5* locus was present in ~75% of the reads in the control population. This observation indicates that we selected for modifier alleles in the initial F1 and F2 populations prior to performing our mapping experiment, and therefore may have also missed modifier loci in our analysis. It is, therefore, challenging to make direct comparisons between the mouse and zebrafish modifiers in the absence of further data. However, based on the identification of 5 linkage peaks on 3 different chromosomes in our data, and the absence of *Itgα5* in the identified mouse modifier locus, we suggest that there are multiple genetic modifiers of the *fn1a*/*Fn1* phenotype.

Although we cannot rule out that the variants in the *itgα5* coding sequence may alter Itgα5 conformation or activity, our analyses did not support this possibility. In addition, our knockdown and overexpression experiments suggest that *itgα5* expression levels regulate phenotypic variation in *fn1a* mutants. In support of this possibility, our analysis of *itgα5* mRNA levels found that increased variance in *itgα5* expression is associated with the more severe *fn1a* mutant phenotype. These results are consistent with the modifier we have identified being a non-coding variant that regulates *itgα5* expression, potentially expression variance rather than absolute expression levels. A recent report found that gene expression variance is functionally constrained in human populations [64], suggesting that variance can indeed be regulated by genetic variants. It will be interesting to investigate whether the variance we observed in *itgα5* mRNA levels between individual larvae extrapolates to single cells, as it would likely disrupt collective cell behaviours such as myocardial cell migration and cardiac morphogenesis. Regardless, further experiments are necessary to determine how *itgα5* modifies the *fn1a* mutant phenotype in zebrafish, and to examine whether this process is conserved in mouse.

## Materials and methods

### Ethics statement

All procedures performed on animals were conducted in accordance with the guidelines of the European Parliament Directive 2010/63/EU on the protection of animals used for scientific purposes and have been approved by the Animal Protection Committee (Tierschutzkommission) of the Regierungspräsidium Darmstadt (reference: B2/1218 and B2/9000).

## Zebrafish husbandry and lines

Adult zebrafish were maintained in 3.5 l tanks at a stocking density of 10 zebrafish/l with the following parameters: water temperature: 27–27.5°C; light:dark cycle: 14:10; pH: 7.0–7.5; conductivity: 750–800 µS/cm. The zebrafish were fed with granulated and live food (*Artemia salina*) 3–5 times per day, depending on their age. Health was monitored twice a year.

Zebrafish embryos and larvae were maintained in egg water (0.3 g/l instant ocean (Aquarium Systems, cat. no. 216030), 75 mg/l calcium sulfate dihydrate (Merck, cat. no. C3771), with 0.000025% w/v methylene blue (Merck, cat. no. M9140)). Larvae were raised under standard conditions. Embryonic and early larval development was staged according to the series described in [65].

The following previously published transgenic and mutant lines were used in this study: *Tg(myl7:EGFP)*[twu26] [66], *fn1a*[tl43c] [25], and *fn1b*[ya14Tg] [34]. The *fn1a*[bns692] mutant allele was generated for this study. All lines were maintained by backcrossing to WT AB zebrafish.

To help keep track of the various matings performed, we refer to matings between animals with the same heterozygous genotype as intercrosses and matings between animals with the same homozygous genotype as incrosses.

## Generation of the *fn1a*[bns692] allele

CRISPR/Cas9 target sites were designed targeting *fn1a* (ENSDARG00000019815; ENSDART00000124346.4) from the GRCz11 assembly using CHOPCHOP [67]. Guide RNAs (gRNAs) were synthesised as described in [68] with Cas9 mRNA produced as described in [69]. 1 nl of an injection mix containing 100 ng/µl of Cas9 mRNA, 50 ng/µl gRNA, and 0.1% w/v phenol red (Merck, cat. no. P0290) in nuclease-free water was injected into 1-cell stage zebrafish embryos. gRNA efficiency was determined using a T7 endonuclease I assay [70], briefly, the genomic target was amplified by PCR. Purified PCR product was denatured and reannealed, then incubated with T7 Endonuclease I (New England Biolabs, cat. no. M0302L) for 60 minutes at 37°C. DNA was then run on a gel to estimate cutting efficiency using the formula: Efficiency = $100 \times (1 - (1 - \text{fraction cleaved})^{\frac{1}{2}})$ as described in [71].

Two guides, one upstream of the transcription start site (target sequence: CTAGATGCATCAAGGTTCGGGGG; cutting efficiency = 26.8%) and the other in intron 17 (target sequence: GATGCTAACAAATGTCCATGTGG; cutting efficiency = 13.1%), were selected based on cutting efficiency. 1 nl of an injection mix containing 100 ng/µl of Cas9 mRNA, 50 ng/µl of each gRNA, and 0.1% w/v phenol red in nuclease-free water was injected into 1-cell stage zebrafish embryos. Embryos were screened for the presence of a large deletion using a primer pair flanking the target sites. Clutches where the deletion was present were raised to adulthood. Germline transmitting founder zebrafish were identified by incrossing and screening pooled embryos for the deletion by PCR. Identified founders were then outcrossed and F1 adults screened by PCR for the presence of the deletion. A single positive F1 zebrafish was chosen at random and outcrossed to establish the *fn1a*[bns692] allele.

## *In situ* hybridisation

*In situ* hybridisation was performed as previously described [72] using published *myl7* [20] and *fn1b* [39] probes. *myl7* expression was analysed in four clutches for each *fn1a* allele. *fn1b* expression was analysed in two clutches for the *fn1a*[bns692] allele. Following post-staining fixation and dehydration, samples were cleared in BABB solution (a 2:1 ratio of benzyl benzoate:benzyl alcohol) and imaged using a Zeiss Axio Imager A2 widefield microscope. All images were processed using Fiji [73]. Approximately 48 embryos, depending on the number of embryos available, were imaged from each clutch in a blinded experiment and subsequently collected in a 96-well plate for genotyping.

## Brightfield imaging

Larvae were anaesthetised with 0.01% tricaine and mounted in 3% w/v methyl cellulose (Merck, cat. no. M0512) in egg water. All images were acquired using a Nikon SMZ25 stereo microscope and processed using Fiji.

## Confocal imaging

Larvae were anaesthetised with 0.01% w/v tricaine and mounted in 0.5% w/v low-melting agarose in egg water. All images were acquired using a Zeiss LSM 800 Examiner confocal microscope and processed using Fiji.

## Genotyping

Genomic DNA was extracted from embryos, larvae, and fin clips using a published protocol [74]. In brief, samples were collected in PCR tubes or 96-well plates, egg water was removed using a pipette, and 50 µl DNA extraction buffer (50 mM KCl, 10 mM Tris pH8, 1 mM EDTA, 0.3% IGEPAL, 0.3% Tween20, 500 µg/ml proteinase K) added. Samples were then incubated at 55°C for 2 hours followed by 99°C for 5 minutes to deactivate the proteinase K. 1 µl of this solution was used in all genotyping assays.

The *natter/fn1a*$^{tl43c}$ allele was genotyped using a custom KASP assay (LGC genomics, aliquot ID: 1217782728). The *fn1a*$^{bns692}$ allele was genotyped by PCR using one forward primer (5'-AACATCTGGCCTGGGAAT-3') and two reverse primers (5'-AGCACTTAGACGCTTTTGC-3'; 5'-AGTGGTTGATTACCCGCC-3') in a single reaction. The 455 bp WT and 529 bp mutant alleles were subsequently separated by gel electrophoresis. The *fn1b*$^{ya14Tg}$ allele was genotyped by PCR using forward (5'-GGAGCGTTGCTATGATGACTCACTGG-3') and reverse (5'-GCTACAAAGTCATGAAAGAGGAATG-3') primers. The 243 bp WT and 310 bp mutant alleles were subsequently separated by gel electrophoresis.

## Single larva RNA/DNA extraction

Single larvae were anaesthetised and collected at the desired stage in 1.5 ml microcentrifuge tubes. Egg water was removed using a pipette and 100 µl TRIzol reagent (Thermo Fisher Scientific, cat. no. 15596018) added per tube along with a small number of 0.5 mm RNase free homogeniser beads (Next Advance, cat. no. ZrOB05-RNA). These tubes were immediately placed in a Bullet Blender homogeniser (Centaur, cat. no. BB24-Au) for 3 minutes at speed 8, after which time the tubes were stored at -80°C prior to extraction. To extract RNA and DNA, samples were first centrifuged at 12000 xg for 5 minutes at 4°C and the supernatant transferred to a new tube leaving behind the homogeniser beads. 20 µl chloroform was then added to each sample and vortexed briefly followed by a 3 minute incubation at room temperature. Samples were centrifuged at 12000 xg for 15 minutes at 4°C and the aqueous phase transferred to a new tube for RNA purification. RNA was purified from the aqueous phase using the RNA Clean & Concentrator-5 kit (Zymo research, cat. no. R016) with on-column DNase digestion according to manufacturer's protocol. To purify DNA, 30 µl of 100% non-denatured ethanol was added to the remaining inter- and organic phases to precipitate the DNA and washes performed according to manufacturer's protocol. In brief, two washes were performed with 0.1 M sodium citrate in 10% v/v non-denatured ethanol, pH 8.5, and one wash was performed with 75% v/v non-denatured ethanol. After centrifugation, ethanol was evaporated by drying at 37°C for 5–30 minutes and the DNA resuspended in 20 µl of 8 mM NaOH.

## RT-qPCR

8 µl of RNA from a 10 µl eluate was reverse transcribed using the Maxima First Strand cDNA Synthesis Kit (Thermo Fisher Scientific, cat. no. K1672). For each clutch, one heterozygous sample was split in two to be used as a no reverse transcriptase (RT) control for genomic DNA contamination whereby one half of the original sample was reverse transcribed as normal while the other half was treated identically with the exception that no reverse transcriptase enzyme was added. These no RT controls were analysed in parallel to empirically determine the detection limit for our experiments. 0.5 µl of a 1:10 dilution of the resulting cDNA samples was used in 10 µl reactions with DyNAmo Colour Flash SYBR Green (Thermo Fisher Scientific, cat. no. F416XL) and 0.25 µM forward and reverse primers in a QuantStudio 7 Pro RealTime thermocycler (Thermo Fisher Scientific). All reactions were performed in technical triplicate and a minimum of three biological replicates were examined in all experiments. Cq values, the PCR cycle where the sample's reaction curve crosses the background threshold level, were automatically determined using the Design & Analysis Software (Thermo Fisher

Scientific, v2.8) with default settings. The amplification efficiency of all primer pairs was determined in separate experiments. Expression was normalised to *rpl13a* in each sample using equations described in [75] with custom R scripts. The variance in *itga5* expression was calculated in R using the var() function.

The following primer sequences were used to amplify target genes: *fn1a* RT-qPCR F1 – 5'-GTGCAGTGTATGCC GAAAG-3', *fn1a* RT-qPCR R1 – 5'-TGTCCGTCCATAACACATCC-3'; *fn1a* RT-qPCR F2 – 5'-CCACCCACTGATC TGAACCT-3', *fn1a* RT-qPCR R2 – 5'-TCACTCTGTAGCCCGTGATG-3'; *fn1a* RT-qPCR F3 – 5'-AGAACCTTCGC GTCCATC-3', *fn1a* RT-qPCR R3 – 5'-AGGGGGAAGCTGCTCAAT-3'; *fn1b* RT-qPCR F – 5'-CAGGGGTCAAGTG TAGATCC-3', *fn1b* RT-qPCR R – 5'-TATCCTTTGGTCGCTCGTAG-3'; *itga5* F – 5'-ACGGCAGAGCTCATTGAA-3', *itga5* R – 5'-AGTTTCAGGTCAGGGACG-3'; *rpl13a* RT-qPCR F – 5'-TGGAGGACTGTAAGAGGTATGC-3', *rpl13a* RT-qPCR R – 5'-ACGCACAATCTTGAGAGCAG-3'. *rpl13a* primers are described in [76], and *fn1a* primer pair 2 in [77].

## mRNA overexpression

*fn1a* and *fn1b* sequences from WT cDNA obtained from mixed staged AB embryos were cloned into the pCS2 + expression vector using NEBuilder HiFi DNA Assembly Master Mix (New England Biolabs, cat. no. E2621S). The *itga5* coding sequence was previously cloned into the pCS2 + plasmid [51]. mRNA was transcribed from NotI linearised plasmid templates using the mMESSAGE mMACHINE SP6 Transcription Kit (Thermo Fisher Scientific, cat. no. AM1340) and purified with Phenol:Chloroform:Isoamyl Alcohol (Thermo Fisher Scientific, cat.no. 15593031) followed by isopropanol precipitation according to manufacturer's protocol. Purified mRNA was aliquoted and stored at -80°C. For injections, an aliquot of mRNA was defrosted on ice, and an injection mix prepared with a final concentration of 250 or 350 ng/μl mRNA and 0.1% w/v phenol red in nuclease-free water. 1 nl of this injection mix was injected into 1-cell stage zebrafish embryos.

## Temperature shift experiments

Embryos were collected in petri dishes with egg water 15 minutes after removing dividers from mating tanks. Embryos were allowed to develop at 28.5°C until mid-blastula stages before sorting. Up to 60 fertilised embryos were placed in each of three clean petri dishes with fresh, pre-heated egg water and allowed to continue developing at 28.5°C until the bud stage (10 hpf). Embryos were sorted at the bud stage to remove embryos with defects as well as those displaying asynchronous development. One dish from each clutch was then transferred to a 32°C incubator, another dish returned to the 28.5°C incubator; and the remaining dish allowed to develop outside the incubator at approximately 22°C. Dishes were gently agitated at regular intervals for 30 minutes following this transfer to ensure even temperature throughout the dish. Depending on the experiment, embryos were allowed to develop until they reached the protruding mouth stage (72 hpf when raised at 28.5°C), or the equivalent of a 100 hpf larvae when raised at 28.5°C. Once they reached the desired stage, larvae were collected for RNA/DNA extraction, or imaged and collected for genotyping.

## Generating *fn1a* mutant adults

To generate the F1 populations of mutants for the *natter/tl43c* and *bns692 fn1a* alleles, heterozygous zebrafish were intercrossed and the embryos raised at approximately 22°C until they inflated their swim bladder. Larvae were then sorted and the WT looking larvae raised under standard conditions. F1 adults were fin clipped and genotyped to identify mutants. The F2 and subsequent generations (with the exception of the *natter/fn1a^{tl43c}* F3 [32°C] generation) were generated by incrossing mutant adults and raising the progeny at 28.5°C under standard conditions until they reached 100 hpf, when they were phenotyped and the WT looking larvae selected for raising as before. The *natter* F3 [32°C] generation was generated by temperature shifting F3 larvae at 32°C from the bud stage until they inflated their swim bladder, then sorting and raising WT looking larvae as before.

## Sample collection for modifier mapping

A single mating pair of *natter/fn1a^tl43c^* F2 mutants were incrossed and the resulting progeny raised at approximately 22°C until they had inflated their swim bladder. The WT looking larvae were selected and raised under standard conditions to generate an F3 mapping population consisting of 18 adult zebrafish. This population was separated by sex and placed in groups of 3–4 zebrafish resulting in two groups of males and three groups of females. Each group was housed separately over the course of the experiment. A round-robin breeding scheme was used whereby each group of males was set up with each group of females in multi-pair matings over three successive weeks to ensure that the embryos were collected from as many different animals as possible. F3 matings were repeated at weekly intervals and all progeny raised at 28.5°C. Embryos were collected in petri dishes with egg water 15 minutes after removing dividers and immediately transferred to a 28.5°C incubator. At mid-blastula stages, up to 60 fertilised embryos were placed in each petri dish with fresh, pre-heated egg water. As many fertilised embryos as possible were collected from each clutch. At the bud stage (10 hpf), approximately 100 embryos were collected from each clutch to estimate allele frequencies in the F3 mutant population. The remaining embryos were allowed to develop until 100 hpf when mutants displaying the WT-like phenotype were collected to examine which variants were enriched in this population. After collection of each population in a 1.5 ml microcentrifuge tube, egg water was removed with a pipette and the collected tubes immediately frozen at -80°C. In total, 687 embryos at 10 hpf and 97 larvae displaying the WT-like phenotype at 100 hpf were collected from a total of 2,520 fertilised embryos obtained from 6 clutches. After all samples had been collected, they were defrosted and homogenised by hand with sterile plastic pestles (Axygen, cat. no. PES-15-B-SI). Genomic DNA was extracted from homogenised samples using the QIAamp DNA mini kit (Qiagen, cat. no. 51304) according to manufacturer's protocol and the concentrations determined using a Qubit 4 Fluorometer (Thermo Fisher Scientific, cat. no. Q33226).

## Whole-genome sequencing and linkage analysis

Genomic DNA from each clutch was pooled separately in equal amounts for both 10 hpf control embryos as well as 100 hpf WT looking mutant larvae. The resulting pooled samples were sequenced on an Illumina NovaSeq 6000 with 2x 150 bp paired-end reads by CeGaT (Tübingen, Germany). Sequencing coverage was 73.59x and 77.77x for 10 hpf control and 100 hpf WT-like samples, respectively.

Sequenced raw reads were analysed using the WheresWalker mapping pipeline. In brief, POLCA from the MaSuRCA toolkit was used to align the reads to GRCz11 and to identify variants [78]. The resulting VCF files were inputted to WheresWalker [48], which calculates a SNP index to quantify the relative heterozygosity between the two samples such that the genomic regions that are more homozygous in mutant larvae displaying the WT-like phenotype have an elevated SNP index. The linkage peak on chromosome 23 between 29.69 - 36.77 Mbp had the highest SNP index and was selected by the pipeline for further analysis. Additional lower intensity peaks were identified by examining the genome-wide SNP index. Variants were retrieved from the VCF files and annotated using Ensembl Variant Effect Predictor [49]. Variants were filtered based on their predicted consequence to identify non-synonymous coding variants in *itga5*, and the SIFT score [50] used to interpret their potential impact on protein function.

## Morpholino injection

A published translation blocking *itga5* morpholino oligonucleotide (MO) (5'-TAACCGATGTATCAAAATCCACTGC-3') [51] was ordered from Gene Tools (Philomath, U.S.A.), resuspended in nuclease-free water, and aliquoted at 10 ng/nl (1.191 mM/l). Prior to injection, the desired number of aliquots were defrosted, heated at 65°C for 5 minutes, and an injection mix prepared with a final concentration of 1 or 5 ng/nl MO and 0.1% w/v phenol red in nuclease-free water. 1 nl of this mix was injected into 1-cell stage embryos.

## Statistics

All statistical analyses were performed in R [79] using custom scripts with the following packages: tidyverse, ggtext, shadowtext, ggsignif, viridisLite, broom, and openxlsx.

A chi-squared test was used to test the association between genotype and phenotype in each clutch in Figs 1B, 1E, 2C and 2D; the association between phenotype and the injection condition in each clutch in Figs 3, 4C and S3A-C; the association between phenotype and developmental temperature in each clutch in S4A and S4B Fig; the association between phenotype and filial generation in Fig 4A' and 4A''; and the association between phenotype and the age of the zebrafish at spawning in S5 Fig.

Fisher's exact test with Bonferroni correction for multiple comparisons was used to test the association between genotype and phenotype in Fig 3E.

One-way ANOVA with Tukey's post-hoc test for multiple comparisons was used to test the association between genotype and *fn1a* expression levels in S2A–C, S2G–I, S3D, S3E, S4C–H Figs and S1 Table; and the association between mutant phenotype and *fn1b* expression levels in Fig 3A.

A Wilcoxon rank-sum test was used to test the association between mutant phenotype and *fn1b* expression levels in S3F Fig.

The significance level, α, was defined as 0.05 in all experiments. A result was considered to be statistically significant when $p \leq \alpha$.

## Use of artificial intelligence tools and technologies

ChatGPT-4o (OpenAI, San Francisco, U.S.A.) was used for writing the code used in this study and for editing the manuscript.

## Supporting information

**S1 Fig. The *fn1a^bns692^* allele has a large deletion removing the proximal promoter and the first 17 exons of *fn1a*.** **(A)** Schematic showing the *fn1a^bns692^* 27,664 bp deletion. Exons (n = 46) are indicated by black boxes. The 5' and 3' UTRs are indicated by open boxes. The position of the deletion as well as of the PCR primers used to genotype *fn1a^bns692^* mutants (blue), to test for the possible reintegration of the deleted region (red), and for the RT-qPCR analysis (green) is indicated. **(B)** Genotyping of *fn1a^bns692^* homozygous WT, heterozygous, and mutant embryos by PCR. **(C)** Three primer pairs were used to test for the possible reintegration of the deleted region by PCR. GeneRuler 100 bp DNA ladder is present in all gels.
(TIF)

**S2 Fig. The *fn1a^bns692^* allele is not an RNA-less allele.** *fn1a* mRNA levels were examined in 72 hpf larvae with three sets of RT-qPCR primers (see annotation in S1 Fig). Each data point represents an individual larva. Expression is relative to homozygous WT siblings in each clutch. Two clutches were collected from single pair *fn1a^bns692^* heterozygous intercrosses in **(A-F)**, and single pair *natter/fn1a^tl43c^* heterozygous intercrosses in **(G-L)**. All primer pairs reveal an almost complete loss of *fn1a* expression in *fn1a^bns692^* mutants **(A-C)** when compared with WT siblings and with *natter/fn1a^tl43c^* mutants **(G-I)**. However, the Cq values indicate that low mRNA levels are detectable from the 3' *fn1a* sequence in *fn1a^bns692^* mutants – compare **(D)** with **(E)** and **(F)**, whereas Cq values are consistent for all three primer pairs used in *natter/fn1a^tl43c^* mutants **(J-L)**. All p-values were calculated using one-way ANOVA with Tukey's post-hoc test for multiple comparisons.
(TIF)

**S3 Fig. *fn1a* and *fn1b* mRNA injections rescue the *fn1a* mutant cardiac phenotype.** Quantification of the 100 hpf cardiac phenotype in uninjected and *fn1a* (A), or *fn1b* (B) mRNA injected mutant larvae from five single pair *natter/fn1a^tl43c^* mutant incrosses. (C) Quantification of the 100 hpf cardiac phenotype in uninjected and *fn1a* mRNA injected mutant larvae

from five single pair *fn1a*[bns692] mutant incrosses. The numbers on each bar indicate the total number of larvae assessed. *fn1b* mRNA levels were analysed by RT-qPCR in uninjected (D) and 350 pg *fn1a* mRNA injected (E) 72 hpf *fn1a*[bns692] mutant larvae and WT siblings collected from two single pair heterozygous intercrosses. Two clutches were collected and analysed. Each data point represents an individual larva. Expression is relative to homozygous WT siblings in each clutch. (F) *fn1b* mRNA levels were analysed by RT-qPCR in the hearts and bodies of dissected 72 hpf *fn1a*[bns692] mutant larvae collected from single pair mutant incrosses. Each data point represents pooled tissue; 5–10 larvae were dissected for each pool. Mutants displaying the WT-like and mild phenotypes were pooled together. Severe mutants were pooled separately. Expression is relative to the WT-like/mild mutant pool in each clutch. For (A-C), p-values were calculated using a chi-squared test; for (D-E), p-values were calculated by one-way ANOVA with Tukey's post-hoc test for multiple comparisons; and for (F), p-values were calculated by a Wilcoxon rank-sum test.
(TIF)

**S4 Fig. The *fn1a* mutant cardiac phenotype is temperature dependent, but temperature does not alter the relative expression of *fn1a*.** Quantification of the 100 hpf cardiac phenotype in *natter/fn1a*[tl43c] **(A)** and *fn1a*[bns692] **(B)** mutants from three single pair heterozygous intercrosses. Each clutch was split into three dishes at an early blastula stage and raised at 28.5°C until the embryos reached the bud stage (10 hpf). Dishes were then incubated at different temperatures until the larvae reached the equivalent of a 100 hpf larva raised at 28.5°C, and were then phenotyped. The numbers on each bar indicate the total number of mutant larvae assessed. *fn1a* mRNA levels were examined at the protruding mouth stage (72 hpf when raised at 28.5°C) in two clutches from single pair *natter/fn1a*[tl43c] **(C-E)** and *fn1a*[bns692] **(F-H)** heterozygous intercrosses. Each clutch was split into three dishes at an early blastula stage and raised at 28.5°C until they reached the bud stage (10 hpf). Dishes were then incubated at 22°C **(C, F)**, 28.5°C **(D, G)**, or 32°C **(E, H)** until they reached the protruding mouth stage, and were then collected for RT-qPCR analysis. Each data point represents an individual larva. Expression is relative to homozygous WT siblings in each clutch. p-values were calculated using a chi-squared test in **(A)** and **(B)**, and one-way ANOVA with Tukey's post-hoc test for multiple comparisons for **(C-H)**.
(TIF)

**S5 Fig. The *natter/fn1a*[tl43c] mutant cardiac phenotype varies with parental age.** Single pair F3 [28.5°C] **(A)** and F3 [32°C] **(B)** *natter/fn1a*[tl43c] mutants were repeatedly incrossed over multiple weeks and the resulting larvae phenotyped at 100 hpf. The numbers on each bar indicate the total number of larvae assessed. All p-values were calculated using a chi-squared test.
(TIF)

**S6 Fig. An *itga5* MO leads to dose dependent somite, cardiac, and craniofacial defects. (A)** Images of 100 hpf uninjected, 1 ng *itga5* MO injected, and 5 ng *itga5* MO injected larvae from single pair WT incrosses are shown. Outlined regions in **(A)** are enlarged in **(B)** to show the somite phenotype. While the majority of 1 ng *itga5* MO injected larvae resembled the uninjected control, a small proportion developed U-shaped somite boundaries (white arrows in **(B)**) as pictured. 5 ng *itga5* MO injected larvae failed to inflate their swim bladder, developed pericardial oedema and craniofacial defects (white arrowheads in **(A)**), and displayed defects in somitogenesis. Lateral views are shown with anterior to the left. The proportion of larvae matching the image shown is indicated in the top right corner of each image. Scale bars, 1 mm.
(TIF)

**S1 Table. Pairwise comparison of relative *fn1a* mRNA levels in *fn1a* mutants raised at different temperatures.**
(XLSX)

**S2 Table. Annotated list of all variants identified in the chromosome 23:31.29 - 35.29 Mbp linkage region.**
(XLSX)

**S3 Table. List of all oligonucleotide sequences used in this study.**
(XLSX)

**S4 Table. Phenotyping counts for all figures.**
(XLSX)

**S5 Table. Cq values for all samples.**
(XLSX)

## Acknowledgments

We thank Richard Hynes and Sophie Astrof for discussion at early stages of this project, Douglas Adamoski Meira and Lara Falcucci for comments on the manuscript, and Carmen Büttner, Denis Grabski, Hans-Martin Maischein, Simon Perathoner, and all of the zebrafish facility staff for technical support.

## Author contributions

**Conceptualization:** Samuel J. Capon, Didier Y.R. Stainier.

**Formal analysis:** Samuel J. Capon, McKenna Feltes.

**Funding acquisition:** Scott A. Holley, Steven A. Farber, Didier Y.R. Stainier.

**Investigation:** Samuel J. Capon, Anastasia Maroufidou, Yanli Xu, Darpan Kaur Matharoo.

**Project administration:** Didier Y.R. Stainier.

**Resources:** McKenna Feltes, Dörthe Jülich, Scott A. Holley, Steven A. Farber, Didier Y.R. Stainier.

**Supervision:** Didier Y.R. Stainier.

**Writing – original draft:** Samuel J. Capon, Didier Y.R. Stainier.

**Writing – review & editing:** Samuel J. Capon, Anastasia Maroufidou, McKenna Feltes, Yanli Xu, Darpan Kaur Matharoo, Dörthe Jülich, Scott A. Holley, Steven A. Farber, Didier Y.R. Stainier.

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
