## [Decision Letter · Decision Letter 0]

PGENETICS-D-25-00099

A genetic modifier links integrin α5 to the phenotypic variability of fibronectin 1a mutant zebrafish

PLOS Genetics

Dear Dr. Stainier,

Thank you for submitting your manuscript to PLOS Genetics. After careful consideration, we feel that it has merit but does not fully meet PLOS Genetics's publication criteria as it currently stands. Therefore, we invite you to submit a revised version of the manuscript that addresses the points raised during the review process.

Please submit your revised manuscript within 30 days Apr 12 2025 11:59PM. If you will need more time than this to complete your revisions, please reply to this message or contact the journal office at plosgenetics@plos.org. Please include the following items when submitting your revised manuscript:

We look forward to receiving your revised manuscript.

Kind regards,

Ophir Klein

Academic Editor

PLOS Genetics

Fengwei Yu

Section Editor

PLOS Genetics

Aimée Dudley

Editor-in-Chief

PLOS Genetics

Anne Goriely

Editor-in-Chief

PLOS Genetics

**Journal Requirements:**

3) Please ensure that the supplementary Figures are uploaded separately with the file type 'Supporting Information'. Please ensure that each Supporting Information file has a legend listed in the manuscript after the references list.

Potential Copyright Issues:

i) Please confirm (a) that you are the photographer of 1, and S6, or (b) provide written permission from the photographer to publish the photo(s) under our CC BY 4.0 license.

5) Please ensure that the funders and grant numbers match between the Financial Disclosure field and the Funding Information tab in your submission form. Note that the funders must be provided in the same order in both places as well. Currently, "the Max Planck Society" is missing from the Funding Information tab.

**Reviewers' comments:**

Reviewer's Responses to Questions

Reviewer #1: The manuscript from Capon et al examines genetic modifiers of zebrafish fibronectin 1a (fn1a) mutants. Understanding genetic modifiers can help understand the etiology of diseases. However, we lack considerable understanding of genetic modifiers that may positively or negatively influence disease in animal models as well as humans. It is long been understood that the fn1a/natter mutants have significant variability in penetrance and expression of their cardiac defects with some mutant hearts looking almost WT. The study creates a new fn1a allele and shows that phenotypic variability is not unique to the natter allele and is also observed with the new allele, a large deletion compared with the missense mutation of natter. The authors confirm that increased temperature and show that parental age both can lead to increased penetrance of severe cardiac defects in fn1a mutations. While fn1b, a zebrafish fn1a paralog, has increased expression associated with the sever cardiac defects, it’s mRNA overexpression instead rescues the cardiac defects similar to fn1a mRNA overexpression. As effects on fn1b are likely not the cause of increase severity in fn1a mutants, the authors find they can select for carriers that have more WT-like hearts. They then perform genetic mapping to identify different modifiers linked with better phenotypic outcome (WT-like in mutants). They find itga5, a known interactor and mediator of fn1a, has mutations that are linked with WT-like mutants. Depleting itga5 is sufficient to increase the severity of the cardiac defects in fn1a mutants. Thus, they conclude that decreased itga5 expression contributes to variability found in fn1a mutants.

Many experiments of the manuscript are of high quality and the experiments overall support the authors’ conclusions. Work of this type is a step in the right direction for understanding phenotypic variability in animal models and the nature of cardiac defects. However, there are a few areas that could strengthen the manuscript.

Major issues:

1. The itga5 mutations identified were linked to the WT-like mutant phenotype. While depleting itga5 is sufficient to worsen the cardiac defects, the authors fail to show that itga5 expression is affected in the mutants with WT-like cardiac defects.

a. Was the expression of itga5 transcripts checked in the WT-like and severe mutants compared to WT controls like for fn1b in Figure 3A? Wouldn’t one predict higher levels of itga5 expression in the WT-like mutants? The authors may already have samples they are able to perform the analysis with. If expression is affected, would this suggest a SNP in a non-coding region that affects expression?

b. Is itga5 mRNA overexpression sufficient to rescue or reduce the penetrance of the severe defects similar to fn1a or -b mRNA overexpression? Is there an alteration in the ability of itga5 mRNA with the linked mutations that affect the coding sequence to rescue the fn1a mutant cardiac defects? Just because the mutations are predicted to not impact function does not mean that is the case.

2. While both fn1a alleles show temperature sensitivity as stated on line 247, they are not exactly the same. Natter mutants show more sensitivity from 28.5C to 32C, while the deletion allele shows sensitivity between 22C and 28.5C. This may be a minor point. However, it is something that should more explicitly be acknowledged in the manuscript.

Minor issues:

1. In Figure 3C and 3B, standard in situ is not really quantitative. While it is understood that the data shown fit with the quantification inf Figure 3A, use of a more quantitative in situ method, such as HCR or RNAscope, may allow for a better illustration of this observation.

2. Lines 167-169 indicate that they used natter mutants, indicating they can raise natter mutants. Can you please reference that these mutants with WT-like hearts can be raised? If this has not been shown previously, this statement and use of natter mutants needs a little more background for the reader. It is not clear this can be done until later in the manuscript.

Reviewer #2: In this manuscript, Capon and colleagues investigate the fascinating topic of the genetic basis for phenotypic variability. This is an important and timely area of study, with potential relevance to our comprehension of the distinct outcomes seen for patients with comparable genotypes. To make progress toward this level of comprehension, the field needs more concrete examples of genetic modifiers that underlie phenotypic variation. Here, the authors have selected an interesting and convenient scenario for the study of variation: as shown by prior studies, mutation of the zebrafish fn1a gene results in phenotypes that are lethal when severe and undetectable at their most mild. Through a series of cleverly designed and clearly presented experiments, they demonstrate several contributors to variation in zebrafish fn1a mutants — temperature, an age-dependent parental factor, and genetic modifiers — and they present evidence supporting the conclusion that a genetic modifier in the itga5 locus contributes to the phenotypic variability in fn1a mutants. This interesting set of analyses will be of high interest to readers interesting in the genetic basis for variability, and it seems likely to inspire other studies of this type that would be focused on other loci. Value to readers could be enhanced by a few additions or clarifications. Specifically:

1) Have the authors examined whether itga5 expression levels are higher or lower in fn1a mutants with mild or severe phenotypes? An experiment parallel to that shown in Fig. 3A for fn1b would be helpful, even if the results do not show a correlation between itga5 expression levels and phenotypic severity.

2) Along similar lines, does heterozygosity for a mutation in itga5 modify the fn1a mutant phenotype?

3) The authors have focused their attention on scoring variability in the cardiac phenotype in fn1a mutants. This is a reasonable choice, although readers will likely be interested in how the other aspects of the fn1a mutant phenotype, such as the disorganization of skeletal muscle, are modified by temperature, parental factors, and itga5. While other phenotypic features may not have been monitored closely, any available information could be valuable to provide.

4) Could the authors clarify the significance of the “symmetric” and “asymmetric” cardiac phenotypes, as highlighted in Fig. 1A,B? It seems to be implied that symmetric is the “more normal” scenario. What do the authors think that the cardiac cells are doing in the asymmetric scenario?

5) When discussing the basis for the temperature-dependent variation seen in fn1a mutants, would the authors want to consider making reference to a PLoS One article by Lin et al. (2013), “Low temperature mitigates cardiac bifida in zebrafish embryos”? This article seems to make the case that multiple cardia bifida phenotypes (caused by mutations in s1pr2, gata5, or bon) are modified by temperature. Perhaps these variable phenotypes also relate to an impact on fn1a, or perhaps the process of cardiomyocyte migration is inherently temperature-dependent?

Reviewer #3: In this manuscript by Capon and colleagues, the authors address the important topic of phenotypic variation often observed among individuals with the same genetic mutation. This is a broad topic with very high importance to all researchers in the field. The manuscript is rigorous in that they rule out several hypotheses before using WGS to identify a putative modifier of the fn1a phenotype. They follow up on this sequencing hit with morpholino induced knockdown demonstrating that downregulating itga5 can phenocopy the severe condition. In addition to this partial mechanism of variation, they also observe parental age effects, temperature sensitivity, and unexpected paralog upregulation that correlates with more severe phenotypes. As such, a large portion of this paper is “phenomenological” rather than mechanistic, which is just fine with this reviewer. Especially because the phenomena appear to be rigorously demonstrated and very interesting. For the more mechanistic portion, the modifier mapping to the interval containing the itga5 locus has a few issues that need to be addressed which are detailed below. Most importantly, are the putatively causative modifier SNP alleles that are reported to rescue the phenotype ever detected in the severe individuals? Overall this is an exciting paper, with some mechanistic insight as well as other more preliminary fascinating observations that will lead to follow up studies in the future outside the scope of the current work.

fn1b upregulation:

The authors make the surprising and interesting finding that the fn1b paralog is upregulated in severe fn1a mutants compared with mild fn1a mutants. Then go on to show that this upregulation is not causative of the phenotypic differences. This is one of many examples of high rigor in this paper ruling out various hypotheses which is important.

The authors state on line 208 that : “Notably, we never observed fn1b upregulation in mutant larvae with no phenotype, or even a mild phenotype” However, in the in situ images in figure 3C it appears that one animal in the middle panel may not have overall increased expression, however the pattern appears different compared with the genetic wild type in the left panel. Can the authors clarify if they interpret this to be identical expression compared with the genetic wild type? I’m assuming this is a mild mutant and not a wild type-like, but the expression pattern does appear slightly different. Please clarify.

Although the authors cannot rule out transcriptional adaptation with the deletion allele, it is interesting that it still produces upregulation of fn1b. Additional discussion of potential mechanisms underlying this observation is warranted. How “this upregulation [fn1b] is most likely the result of the loss of Fn1a protein” needs to be further fleshed out. Can the authors clarify further what they think the mechanism is? The authors indicate that fn1b can compensate so upregulation is a consequence of a severe phenotype but don’t suggest a mechanism of upregulation or if there might be a difference in the upregulation seen between the different fn1a mutant alleles. If some other gene in the pathway were disabled (perhaps itga5), indirectly inhibiting fn1a activity, then would the upregulation of fn1b be predicted?

It is a bit confusing that the primers that sit inside the deleted region still amplify product in the deletion animals when these sequences are not present in the genome of these animals. Perhaps I’m not understanding but the authors state “however careful analysis of the cq values associated with primer pairs inside and outside the deleted region detected low levels of transcript from intact exons.” Are the levels of genetic material in primer pair #1 (inside deletion) simply noise, while the other pairs are detectable RNA? Could the low levels of transcript from primer pairs within the deletion be detection of maternally deposited wild type mRNA that is being detected? Please clarify.

It is good that the authors speculate why the fn1a protein would be predicted to be nonfunctional from the deletion, even if some protein is produced.

Transcriptional adaptation

I applaud the authors for generating an allele that is not likely to undergo NMD and induce TA. The authors then determine that this allele is not an RNAless allele as some transcript is detected. However, the levels are significantly lower than wild type. Could this in fact be due to NMD? Even if unlikely, the authors could complete this analysis by blocking NMD to determine if this has any effect on the phenotype and confirm that the downregulation of transcript in their alleles (including the deletion allele) is at least partially due to NMD, in addition to the proximal promoter deletion. This would help to further rule out NMD-induced TA.

Rigorous methodology

The authors formulated and tested numerous hypotheses to understand the fn1a phenotype variation, including a new deletion allele to assess what might be the null condition to complement the initial findings made with the allele recovered in the ‘96 screen.

Temperature sensitivity

“raised at the standard temperature” in the abstract comes out of nowhere and more context is needed for the reader to understand why this is relevant without reading the entire manuscript.

Early on the authors say that the temperature sensitivity might be due to transcript stability but then later say that they don’t see differences in transcript stability. In the same paragraph, they say the temperature sensitivity is due to protein loss but don’t actually test this. The authors could consider some protein detection assay to formally test this hypothesis and confirm or rule out this model.

Selective breeding

Selective breeding is a very powerful way to study phenotypic variation and compare animals that are sensitive or resilient to a given mutation. The selective breeding for mutants with wild type phenotypes is outstanding. However, also selecting and purifying for families that are severe might allow for a better comparison to further understand the mechanisms. More on this topic below.

Modifier mapping

It is very impressive and exciting the authors mapped to regions of the genome that they think underlie phenotype severity. It is remarkable that they had the power to get to a likely causative genomic region that perhaps segregates with severity. However, because the authors compare the wild type like mutants to the unselected background they cannot be sure that the alleles they recovered actually segregate with the wild type like mutant phenotype. To make stronger, the authors could hybridize their selectively bred strain to an unselected background, then resegregate to make sure these SNPs aren’t just inherited by descent but are actually segregating with severity. These analyses would be stronger had the authors compared severe individuals to their selectively bred families rather than comparing with the mixed or background condition.

In this similar vein, are the alleles of the SNPs that they think confer the mild/wild type like phenotype ever detected in severe individuals? This is important because the SNPs may be fully sufficient to confer the mild phenotype, or (more likely) that they are one part of a broader buffering mechanism that may not always segregate with the mild or severe condition.

How can itga5 rescue? Is there a molecular mech? They do speculate that might affect binding or just be expression. This is very speculative but still interesting. This should be fleshed out and more rigorously tested in future studies.

It is strong that they mechanistically test if the changes in itga5 expression are functional with morpholino injection. Since the MO matches the published mutant phenotype and they use a low dose that doesn’t produce heart defects in wild type this approach is ok but this experiment would be stronger with a germline genetic loss of function. Although I understand that a loss of function coding mutant may not be the perfect experiment since (if I understand the model correctly) the authors think the SNP confers itga5 expression differences which may be difficult to recapitulate with a germline transmitting allele that affects expression.

Parental age effects

This is a very surprising and interesting observation. I appreciated and enjoyed looking at the references that indicate a similar phenomenon in the janus mutant which I was not familiar with. It is so striking that penetrance and expressivity of a given mating pair varies from cross to cross becoming more severe as the parents age. I look forward to future work fleshing out the mechanism behind this fascinating observation.

Minor comments

Authors use the terms variability and variation interchangeably. These two terms have different meanings, variability is defined as the propensity to vary, while variation is a measurable statistic as defined by Gunter Wagner 1996 “complex adaptations…evolvability” (maybe his earlier works too).

**Have all data underlying the figures and results presented in the manuscript been provided?**

Reviewer #1: **No: ** It is not clear that the whole genome sequencing data have been deposited. If it is required by the journal, they should deposit the raw data.

Reviewer #2: Yes

Reviewer #3: Yes

PLOS authors have the option to publish the peer review history of their article (what does this mean? ). If published, this will include your full peer review and any attached files.

**Do you want your identity to be public for this peer review?** For information about this choice, including consent withdrawal, please see our Privacy Policy .

Reviewer #1: No

Reviewer #2: No

Reviewer #3: No

**Figure resubmission:**
---

## [Editor Report · Decision Letter 1]

Dear Dr Stainier,

We are pleased to inform you that your manuscript entitled "A genetic modifier links integrin α5 to the phenotypic variation in fibronectin 1a mutant zebrafish" has been editorially accepted for publication in PLOS Genetics. Congratulations!

Yours sincerely,

Ophir Klein

Academic Editor

PLOS Genetics

Fengwei Yu

Section Editor

PLOS Genetics

Aimée Dudley

Editor-in-Chief

PLOS Genetics

Anne Goriely

Editor-in-Chief

PLOS Genetics

Comments from the reviewers (if applicable):

**Data Deposition**

http://datadryad.org/submit?journalID=pgenetics&manu=PGENETICS-D-25-00099R1

**Press Queries**

---

## [Editor Report · Acceptance letter]

PGENETICS-D-25-00099R1

A genetic modifier links integrin α5 to the phenotypic variation in fibronectin 1a mutant zebrafish

Dear Dr Stainier,

We are pleased to inform you that your manuscript entitled "A genetic modifier links integrin α5 to the phenotypic variation in fibronectin 1a mutant zebrafish" has been formally accepted for publication in PLOS Genetics! Your manuscript is now with our production department and you will be notified of the publication date in due course.

With kind regards,

Anita Estes

PLOS Genetics

On behalf of:
